# In Vitro Models of Cardiovascular Calcification

**DOI:** 10.3390/biomedicines12092155

**Published:** 2024-09-23

**Authors:** Andrea Tóth, Enikő Balogh, Viktória Jeney

**Affiliations:** MTA-DE Lendület Vascular Pathophysiology Research Group, Research Centre for Molecular Medicine, Faculty of Medicine, University of Debrecen, 4032 Debrecen, Hungary; andrea.toth@med.unideb.hu (A.T.); balogh.eniko@med.unideb.hu (E.B.)

**Keywords:** vascular calcification, valve calcification, vascular smooth muscle cell calcification, valve interstitial cell calcification, calcification inducers, calcification detection

## Abstract

Cardiovascular calcification, characterized by hydroxyapatite deposition in the arterial wall and heart valves, is associated with high cardiovascular morbidity and mortality. Cardiovascular calcification is a hallmark of aging but is frequently seen in association with chronic diseases, such as chronic kidney disease (CKD), diabetes, dyslipidemia, and hypertension in the younger population as well. Currently, there is no therapeutic approach to prevent or cure cardiovascular calcification. The pathophysiology of cardiovascular calcification is highly complex and involves osteogenic differentiation of various cell types of the cardiovascular system, such as vascular smooth muscle cells and valve interstitial cells. In vitro cellular and ex vivo tissue culture models are simple and useful tools in cardiovascular calcification research. These models contributed largely to the discoveries of the numerous calcification inducers, inhibitors, and molecular mechanisms. In this review, we provide an overview of the in vitro cell culture and the ex vivo tissue culture models applied in the research of cardiovascular calcification.

## 1. Introduction

Calcification of the cardiovascular system, characterized by hydroxyapatite deposition in the arterial wall and heart valves, is associated with high cardiovascular morbidity and mortality. Cardiovascular calcification is a major complication in chronic kidney disease (CKD), diabetes, dyslipidemia, and hypertension and is commonly seen in the aging population [1].

Cardiovascular calcification in young patients is extremely rare and is suspected to be associated with an underlying hereditary monogenic disorder [2]. Early-onset cardiovascular calcification disorders can be classified as diseases caused by an altered extracellular purine/inorganic pyrophosphate (PPi)/inorganic phosphate (Pi) metabolism, interferonopathies, and Gaucher disease [2]. Diseases with altered PPi/Pi metabolism include generalized arterial calcification of infancy (GACI 1, MIM no. 208000; GACI 2, MIM no. 614473), pseudoxanthoma elasticum (PXE, MIM no. 264800), calcifications of joints and arteries (CALJA, MIM no. 211800), idiopathic basal ganglia calcification syndromes caused by heterogenetic mutations in six different genes, and Hutchinson–Gilford progeria syndrome (HGPS, MIM no. 176670) [2]. 

Based on the localization, cardiovascular calcification can be categorized as intimal, medial, and valve calcification; all types have distinct functional consequences, as medial calcification is associated with vessel stiffness and valve calcification causes impairment of aortic valve function [3]. 

Soft tissue calcification was considered a passive process in which Ca-phosphate precipitates spontaneously when the circulating Ca and phosphate levels exceed a threshold. However, detailed research in the previous decades proved that cardiovascular calcification is a highly complex, actively regulated, cell-mediated process [4]. These studies showed that different cell types of the cardiovascular system can undergo a trans-differentiation process, during which they lose their original and gain an osteogenic phenotype in response to certain environmental signals [5,6,7,8]. 

Our goal in this review was to collect all the relevant information about the most frequently used in vitro cellular and ex vivo tissue culture models of cardiovascular calcification. After providing an overview of the different calcification types in the cardiovascular system, we discuss the calcifying cell types and the most commonly used inducers in the in vitro calcification models (Figure 1). We also collected the widely used staining techniques for the detection of calcification. 

## 2. Major Types of Calcifications in the Cardiovascular System

### 2.1. Intimal Calcification

Intimal calcification occurs in the intimal layer of the arterial wall. Intimal calcification is an active process frequently associated with atherosclerosis, characterized by lipid accumulation and inflammation. Atherosclerosis is driven by cardiovascular risk factors, such as aging, diabetes, smoking, and hyperlipidemia [9,10]. For a long time, atherosclerosis has been considered a cholesterol-storage disease, affecting people in high-income countries. This theory was challenged recently, and now we assume that atherosclerosis is an inflammatory disorder that is the leading cause of death globally [11,12]. Inflammation is a major driver in the local, myocardial, and systemic complications of atherosclerosis [11,12]. 

Different calcification patterns have been observed during the progression of atherosclerotic lesions [13]. Atherosclerotic calcification begins with microcalcification, the deposition of small (0.5–15 μm in diameter) calcified particles in the intimal layer [13]. This can progress to punctate (>15 μm to 1 mm in diameter), fragmented (>1 mm), and eventually sheet-like calcification (>3 mm) [13]. There were many debates about whether and how calcification affects plaque stability. Plaque calcification correlates with plaque progression and the narrowing of the vessel [14]. Clinical studies indicated that coronary artery calcium (CAC) score correlates with the possibility of future cardiac events. CAC seems to be a more reliable marker of future adverse cardiovascular events than the Framingham risk index in all studied populations [15]. Surprisingly and somehow contradictory, recent pathologic studies showed that sheet-like calcification is mainly present in stable plaques, while microcalcifications, punctate, and fragmented calcifications are more frequent in unstable lesions [13]. Another type of atherosclerotic calcification is nodular calcification, described as nodular calcium deposits in the luminal surface of the lesion [13]. Besides the size, the location of the calcium deposition is important; for example, microcalcification in the fibrous cap might promote plaque rupture [16]. Overall, these studies suggest that extensive calcification might not directly cause future adverse cardiovascular events, but CAC may reflect the extent of the disease [13]. 

### 2.2. Medial Calcification

Medial calcification, or Mönckeberg’s sclerosis, develops in the tunica media layer of the vessel wall [17]. Medial calcification is mainly associated with age, diabetes, and CKD [18]. Medial arterial calcification is found in 17–42% of type 2 diabetes patients and 27–40% of patients with CKD [19]. In late-stage CKD patients, calcification of the coronary artery (64–77%), the abdominal aorta (49–93%), and the femoral arteries (91%) is widespread [20]. 

The pattern of calcium deposition is substantially different from intimal calcification. In the early stage of the disease, calcification is associated with elastic fibers and is present throughout the medial width [21]. In advanced lesions, circumferential rings of calcium, osteocytes, and bone trabeculae appear in the tunica media [21].

Medial calcification was considered a passive, benign process, but now it is widely recognized that it is an actively regulated process that shares similarities with physiological bone formation [5,21,22]. Clinically, medial calcification triggers vascular stiffness and decreases blood vessel compliance, eventually leading to increased pulse pressure and left ventricular hypertrophy [23]. 

### 2.3. Calciphylaxis

Calciphylaxis, or calcific uremic arteriolopathy, is a rare, devastating, and life-threatening calcification syndrome, mostly associated with CKD or acute kidney injury [23,24]. Calciphylaxis is characterized by the occlusion of microvessels in the dermis caused by calcium deposition, leading to painful skin ulcerations [24,25]. Sepsis, due to the infection of skin lesions, is the major cause of death in patients with calciphylaxis. Calciphylaxis has extremely high mortality with a range of 45–80% [24,25]. Calciphylaxis risk factors include obesity, diabetes, hemodialysis, and female sex [25]. The incidence of calciphylaxis has largely increased in the last ten years among hemodialyzed patients [24]. 

The pathomechanism of calciphylaxis is not completely understood and is still being investigated. Elevation in serum calcium and phosphate levels increases the possibility of subsequent calciphylaxis in patients on dialysis treatment. Furthermore, an overdose of vitamin D and warfarin supplementation may increase the risk of calciphylaxis [25].

### 2.4. Heart Valve Calcification

The cardiac valves provide unidirectional blood flow through both pulmonary and systemic circulation. The most common disease of cardiac valves is calcific aortic valve disease (CAVD), which is the most common indication for heart valve replacement surgeries [26]. CAVD is a chronic disease characterized by thickening, fibrosis, and calcification of the aortic valve leaflets [26]. Based on disease severity, CAVD is divided into aortic valve sclerosis, in which the left ventricular outflow is maintained, and aortic valve stenosis, in which obstruction to the left ventricular outflow is present [26]. According to recent studies, the mechanism of CAVD is similar to that of atherosclerotic intimal calcification [27], involving lipoprotein deposition, chronic inflammation, and actively regulated calcification [28,29,30,31,32,33].

## 3. Osteo-/Chondrogenic Differentiation as the Underlying Cellular Mechanism of Calcification

Without any doubt, calcification is an active and tightly regulated mechanism with the contribution from different cell types in the cardiovascular system. An imbalance between calcification inhibitors and inducers initiates the osteo-/chondrogenic reprogramming of cells, a process in which the cells lose their original phenotype and gain osteoblast- or chondrocyte-like features (Figure 1). 

Elevated Pi [34], calcium [35], bone morphogenetic protein 2 [36], pro-inflammatory cytokines [37,38,39], low-density lipoprotein (LDL) [40,41], lipoprotein(a) [42], glucose [43], advanced glycation end products [44], vitamin D [45], uremic toxins [46], and hypoxia [47,48] are the most important pathophysiological triggers of osteo-/chondrogenic trans-differentiation of VSMCs. Under homeostasis, the calcification inducers are counterbalanced by calcification inhibitors; therefore, the cardiovascular system is protected from abnormal hydroxyapatite deposition. Calcification inhibitors include fetuin-A [49], matrix Gla protein [50], PPi [51], magnesium [52], and iron [53,54]. 

Osteo-/chondrogenic trans-differentiation of VSMCs recapitulates osteogenesis of mesenchymal stem cells in many aspects. Various essential cell signaling pathways, including BMP–Msx2–Wnt [55,56], TGF-β [57], Hedgehog [58], parathyroid hormone [59], fibroblast growth factor 23 [60], Notch [61], and Piezo1 [62], play a critical role in facilitating osteoblast as well as VSMC differentiation. Activation of these osteogenic signaling pathways triggers the upregulation and activation of certain osteoblast- and chondrocyte-specific transcription factors, including Runx2, Sry-related HMG box-9 (Sox9), Msx1 and 2, and osterix, which work together in promoting osteo-/chondrogenic phenotype switch of VSMCs [63]. 

This complex regulatory system was a topic of several recent in-depth reviews [6,8,64,65,66,67,68]. Here we will focus on the frequent cell types and inducers used in in vitro models of cardiovascular calcification. 

## 4. Cell Types Used in In Vitro Models of Cardiovascular Calcification

In the cardiovascular system, several cell types have been characterized with the ability of a spontaneous or induced phenotype switch towards osteoblast-like cells during in vitro culture conditions (Table 1). 

### 4.1. Calcifying Vascular Cells (CVCs)

Watson et al. isolated a subpopulation of cells from human and bovine aorta that form calcified nodules under in vitro cell culture conditions [69]. They named these cells as CVCs. CVCs express bone-specific proteins, such as alkaline phosphatase (ALP), collagen I, and osteocalcin (OCN), and produce hydroxyapatite. Further studies revealed that CVC calcification is a regulated process [40,70,71,72,73] and showed that CVCs originate from local smooth muscle cells and circulating hematopoietic stem cells [74]. 

### 4.2. Vascular Smooth Muscle Cells (VSMCs) 

VSMCs are the major cell types in the vessel wall. Without insult, VSMCs exhibit a contractile phenotype, which is essential for the optimal function of blood vessels, primarily maintaining blood pressure through contraction and relaxation in opposition to the heart [75]. Contractile VSMCs are characterized by a slow proliferation rate and robust expression of a range of contractile proteins, including SM α-actin, SM-22α, SM myosin heavy chains, calponin, and smoothelin [7]. On the other hand, in response to environmental changes, such as metabolic insults or injury, VSMCs can alter their phenotype and gain features of macrophage foam cells, osteoblasts, chondrocytes, or adipocytes and decrease the levels of contractile VSMC markers (Figure 1) [7]. 

Nowadays, human aortic smooth muscle cells (HAoSMCs) are the most widely used cells in investigating the mechanisms of intimal and medial calcifications [76]. Additionally, primary mouse aortic smooth muscle cells isolated from thoracic aortas of mice with different genetic backgrounds and genetic modification are relevant tools to explore new potential mechanisms of vascular calcification [77,78,79]. Besides humans and mice, VSMCs from rats, pigs, and bovines are also used in cellular models of calcification research. The major triggers for the VSMC osteo-/chondrogenic phenotype switch will be discussed in the next chapter. 

### 4.3. Valve interstitial Cells (VICs)

Heart valves are composed of an outer monolayer of valve endothelial cells and several internal layers of VICs [80]. Many lines of evidence suggest that the in situ phenotypic transition of VICs into osteoblast-like cells is involved in valve calcification. First, Johnson et al. established the culture of porcine aortic VICs by isolating leaflets from the aortic annulus of pigs, providing a useful tool to study the cellular basis of valvular heart disease [81]. Mohler et al. identified and characterized calcifying VICs from human and canine aortic valves [82]. Besides these models, VICs obtained from sheep, bovine, and mouse valve leaflets were also used to study the molecular mechanism of valve calcification [83,84,85].

### 4.4. Endothelial Cells (ECs) 

ECs together with SMCs maintain vascular homeostasis. The role of VSMCs in the pathogenesis of vascular calcification is well-established; however, numerous recent studies suggest that ECs also contribute largely to the development of vascular calcification. The role of ECs in vascular calcification was recently reviewed by Jiang et al. [86]. Different factors promote the participation of ECs in the calcification process, including oxidative stress, inflammation, autocrine and paracrine mechanisms, mechanotransduction, and endothelial-to-mesenchymal transition [86,87,88]. The most widely used cell types in these studies are human aorta ECs and umbilical vein ECs. However, in some studies, pulmonary artery ECs, human coronary microvascular ECs, and human skin microvascular ECs were used [89]. Besides the role of vascular ECs in vascular calcification, recent studies revealed the contribution of valve endothelial cells to valve calcification [46,90,91,92]. 

### 4.5. EC–VSMC Co-Culture Systems

Interactions between ECs and VSMCs play a fundamental role in vascular homeostasis. Recent findings suggest that cross-communication between ECs and VSMCs is also important in certain pathological conditions, including vascular remodeling and calcification [93,94]. Recently, a cellular co-culture model has been developed to reproduce the complexity of the vascular niche in vitro, and this system was used to investigate vascular calcification [95]. 

### 4.6. Mesenchymal Stem Cells (MSCs) 

MSCs are multipotent cells, differentiating into bone, cartilage, and adipocyte lineages [96,97]. MSCs are present in the vasculature, and osteo-/chondrogenic differentiation of MSCs is now considered a potential mechanism underlying vascular calcification [73,98]. Cheng et al. showed that Msx2 promotes osteogenic and suppresses adipogenic differentiation of MSCs [99]. The excellent growth potential makes MSCs a useful tool in studies addressing the regulation of osteo-/chondrogenic differentiation [100].

**Table 1 biomedicines-12-02155-t001:** In vitro cell calcification models.

Cell Type	Origin	Major Finding	Reference
Calcifying vascular cell	Human and bovine aorta	Spontaneous increase in osteogenic markers (ALP, OCN) and calcification under in vitro culture conditions.	[69]
Vascular smooth muscle cell	Human aorta	Calcification and gain of osteoblast markers (Runx2, OCN) and loss of smooth muscle markers in response to high Pi.	[34,101]
		Calcification in response to high Ca. Synergistic effect of Ca on Pi-induced calcification.	[35]
		Upregulation of osteogenic markers (Runx2, Sox9, ALP, OCN) and calcification in response to hypoxia (1% O_2_).	[48]
		Hypoxia and a hypoxia-mimetic drug, Daprodustat, enhance Pi-induced calcification.	[47,102,103]
	Bovine aorta	Calcification and increased ALP, osteopontin, and Runx2 in response to BGP-containing osteogenic medium.	[41,104]
		Calcification and increased ALP and osteopontin in response to active vitamin D3.	[45]
		Calcification and gain of osteoblast markers (Runx2, OCN) in response to DEX.	[105]
		Increased osteoblast markers (Runx2, OCN, ALP, and bone morphogenetic protein) in response to high glucose.	[43]
	Mouse aorta	The role of ER stress/ATF4 in calcification.	[77]
		GATA6 accelerates VSMC calcification.	[78]
		Matrix metalloproteinase 3 deletion attenuates osteogenic differentiation.	[79]
Valve interstitial cell	Bovine aortic valve	Calcification of VICs in response to endotoxin and phosphate.	[83]
	Human aortic valve	Characterization of calcifying VICs from aortic valves.	[82]
		Hypoxia and a hypoxia-mimetic drug, Daprodustat, enhance Pi-induced calcification.	[103]
Endothelial cell	Human aorta	Inflammation promotes EC calcification.	[89]
	Ovine mitral valve	Contribution of valve ECs to valve calcification.	[90]
Mesenchymal stem cell	Human aorta	Potential role of MSCs in vascular calcification.	[73,98]
	Mouse embryo	Msx2 regulates osteogenic differentiation of MSCs.	[99]

## 5. Ex Vivo Models of Cardiovascular Calcification

Ex vivo tissue culture models bring many advantageous features to the calcification field. These models provide the complexity of an organ, but the experimental approaches are almost as simple as a cell culture system. Tissues from whole mice as well as cell-specific knock-out mice can be studied to discover novel genes involved in cardiovascular calcification. Additionally, mouse models displaying monogenic mutations that cause cardiovascular calcification phenotypes of human diseases are available [2]. Aorta and valve tissue from these mice can be used in ex vivo experimental settings to expand our understanding of the mechanisms of vascular and valve calcifications. 

### 5.1. Aorta Organ Culture Model

The aorta organ culture model is a powerful and widely used method of investigating the pathophysiology of vasculature in association with several diseases, such as atherosclerosis. Diglio et al. first reported the utilization of rat aortic ring explants to study the growth of neovessels [106]. Akiyoshi et al. established the ex vivo aorta organ culture model of vascular calcification [107]. They dissected the thoracic aorta of C57BL/6 mice and stimulated them with inorganic phosphate (Pi) to induce aorta calcification [107]. We recently used this model to investigate the effect of hypoxia on vascular calcification [48,102]. 

### 5.2. Valve Leaflet Organ Culture Model

Watts et al. established the ex vivo cardiac valve culture model [108]. Human cardiac valves (aortic and pulmonary valves) were isolated postmortem within 24 h of death and cultured for 48 h to determine the viability of valve specimens [108]. Later, an organ culture model of porcine valves was created [109]. The biggest advantages of the porcine model are its availability and the high similarity of the porcine valve to the human valve [109]. Recently, Chester et al. developed a calcification model using intact porcine heart valves stimulated by lipopolysaccharide and Pi [110]. Besides porcine mitral valve leaflets, bovine and sheep mitral valve leaflets have been used to investigate aortic stenosis [111,112]. 

## 6. Most Frequently Used Inducers of In Vitro Calcification and Their Mechanism of Action

Several calcification inducers have been identified in the previous decades, and the molecular mechanism of their pro-calcification effect has been revealed. Here we listed the most frequently used calcification inducers used for in vitro studies (Table 1).

### 6.1. Inorganic Phosphate (Pi)

Based on the correlation between hyperphosphatemia and vascular calcification in CKD patients, Jono et al. tested whether extracellular Pi induces calcification of HAoSMCs [34]. They found that HAoSMCs cultured in media containing physiological levels of Pi (1.4 mmol/L) did not calcify, whereas increased levels of Pi (1.6–2.0 mmol/L) trigger HAoSMC calcification in a dose-dependent manner [34]. Elevated Pi also increased the expression of the osteoblastic differentiation markers, runt-related transcription factor 2 (Runx2) and OCN, and triggered the loss of VSMC lineage markers, SM-22α, and SM α-actin [34,101]. Further studies showed that Pi-induced HAoSMC phenotype switch and calcification depend on Pi uptake through sodium-dependent phosphate co-transporters (PiT-1 and PiT-2) [113,114]. 

### 6.2. Extracellular Calcium 

Hypercalcemia is highly correlated with vascular calcification; therefore, Yang et al. investigated whether increased calcium concentration comparable to levels observed in hypercalcemic individuals induces HAoSMC calcification in vitro [35]. They showed that calcium elevation up to 2.6–3.0 mmol/L under normal Pi conditions (1.4 mmol/L) triggered HAoSMC calcification [35]. High calcium increased phosphate uptake and PiT-1 mRNA expression in HAoSMCs, suggesting that calcium promotes calcification by enhancing the phosphate transport activity of HAoSMCs [35]. Moreover, when both Ca and Pi were used in the osteogenic medium, they synergistically induced HAoSMC calcification [35,115,116]. 

### 6.3. β-Glycerolphosphate (BGP)

BGP serves as a phosphate donor; therefore, it is widely used in in vitro calcification assays alone or in combination with other osteogenic inducers. BGP (10 mmol/L) with insulin (10^−7^ mol/L) and ascorbic acid (50 μg/mL) was first used by Shioi et al. to induce calcification of bovine VSMCs [104]. This calcification mixture upregulated ALP and osteopontin (OPN) [104]. Inhibition of ALP activity inhibited bovine VSMC calcification, showing the functional significance of ALP in BGP-induced calcification [104]. Bear et al. showed that BGP alone enhanced Osterix expression by inducing Smad 1/5/8 activation and Runx2 expression [41]. Alesutan et al. described that in response to BGP, VSMCs increased basal respiration, mitochondrial ATP production, and proton leak and decreased spare respiratory capacity and coupling efficiency. However, they did not modify the glycolytic function or the basal as well as glycolytic proton efflux rate, overall indicating a more oxidative and less glycolytic phenotype [117]. The authors concluded that BGP triggers alterations of mitochondrial function and cellular bioenergetics in VSMCs that may contribute to the phenotypic switch and calcification of the cells [117]. 

### 6.4. Ascorbic Acid (AA)

A cocktail of BGP, AA, and dexamethasone (DEX) is commonly used to trigger osteogenic differentiation of multipotent stem cells. However, the effect of AA on osteogenic differentiation of VSMCs is still controversial. Ciceri et al. showed that AA intensifies Pi-induced VSMC osteoblastic differentiation and calcification [118]. In contrast, Ivanov et al. found that AA inhibits BGP-induced VSMC calcification [119]. AA has been shown to stimulate smooth muscle cell marker expression in VSMCs and control vascular inflammation [120,121].

### 6.5. Vitamin D

Vitamin D plays a crucial role in calcium uptake and bone mineralization. The role of vitamin D and its derivatives in vascular calcification is complex. Although vitamin D is generally beneficial, excessive amounts are associated with medial calcification. Jono et al. showed that 1,25-dihydroxy vitamin D3 (1,25(OH)2D3), an active metabolite of vitamin D3, induces bovine VSMC calcification by inhibiting the expression of a parathyroid hormone-related protein, which is an endogenous inhibitor of vascular calcification [45]. Moreover, they showed that 1,25(OH)2D3 increases ALP activity and OPN expression [45]. Furthermore, studies showed that 1,25(OH)2D3 stimulates VSMC migration and proliferation, contributing to 1,25(OH)2D3-induced vascular pathologies [122,123]. Rajasree et al. found that 1,25(OH)2D3 induces upregulation of the vitamin D receptor (VDR) and increases calcium uptake in rabbit VSMCs [124]. Han et al. described a functional cooperation between VDR and Runx2, which plays a crucial role in 1,25(OH)2D3-induced VSMC calcification [125]. 

### 6.6. Dexamethasone (DEX)

Glucocorticoids, including DEX, promote the growth and osteoblastic differentiation of osteoprogenitor cells and nodule formation [126]. While investigating the effect of DEX on VSMC calcification, Mori et al. showed that DEX enhances several phenotypic osteoblast markers, such as ALP activity and Runx2 expression, and increases the calcification of bovine VSMCs [105]. Additionally, DEX treatment promotes the calcification of vascular pericytes by decreasing the expression of calcification inhibitors, including matrix Gla protein, OPN, and vascular calcification-associated factor [127]. Glucocorticoid receptor (GR) plays a critical role in DEX-induced VSMC calcification, and thus DEX-induced calcification is prevented by GR antagonists [127]. 

### 6.7. High Glucose

Type II diabetes has been associated with an increased prevalence of vascular calcification [128]. Chen et al. showed that high glucose increases the expression of Runx2 and its downstream protein osteocalcin and enhances ALP activity and the secretion of bone morphogenetic protein-2, a known osteo-inductive factor in bovine VSMCs [43]. Additionally, high glucose enhanced BGP-induced VSMC calcification [43]. Elevated glucose levels activate the transcription factor nuclear factor kappa B in VSMCs, which is considered one of the key mediators of the pro-calcific effects of high glucose [129]. 

### 6.8. Hypoxia

Hypoxia is a condition of low oxygen availability, a characteristic feature in all types of cardiovascular calcification. Hypoxia upregulates bone-related markers (Runx2, Sox9, ALP, and OCN) in HAoSMCs and induces calcification [48]. Hypoxia triggers hypoxia-inducible factor (HIF) signaling and mitochondrial dysfunction, leading to the generation of reactive oxygen species (ROS) in HAoSMCs. Balogh et al. showed that the pro-calcification effect of hypoxia is HIF- and ROS-dependent [48]. Additionally, hypoxia and hypoxia-mimetic drugs, including the prolyl hydroxylase inhibitor Daprodustat, enhance high Pi-induced calcification of both HAoSMCs and human VICs in a HIF-dependent manner [47,102,103]. A recent study highlighted the role of endoplasmic reticulum stress and an interplay between HIF-1α and activating transcription factor 4 in Daprodustat-induced promotion of HAoSMC calcification [130]. 

## 7. Staining Techniques to Detect Calcification In Vitro 

Several techniques are currently used to assess biomineralization in cell culture systems using dyes that enable colorimetric (Alizarin red staining and von Kossa staining) or fluorescent (Osteosense and BoneTag) detection of hydroxyapatite. ALP enzymatic activity is also frequently measured to detect osteogenic differentiation. 

### 7.1. Alizarin Red S (ARS) Staining

ARS staining is considered the gold standard for the detection of calcification not only in bone-forming cells but also in cells of the cardiovascular system [131]. The ARS dye is an anthraquinone derivative that forms a stable complex with calcium cations through a chelation process [132]. The ARS–calcium complex is bright red, making it visible to the naked eye. Moreover, quantification of the ARS staining is possible after extracting the complex in hexadecyl pyridinium chloride or 10% acetic acid, followed by spectrophotometric analysis [133].

### 7.2. Von Kossa Staining

Von Kossa stain contains silver ions that react with the different anions, including phosphates or carbonates, in calcium deposits, resulting in a transient yellow coloration [134]. In the second step, the organic material reacts with the bound silver, reducing silver ions to metallic silver in the presence of light or photographic developers. The second reaction yields a dark brown or black precipitate that can be observed with the naked eye [134].

### 7.3. Fluorescently Labeled Probes

OsteoSense is a synthetic near-infrared fluorescent bisphosphonate derivative probe that exhibits rapid and specific binding to hydroxyapatite [135]. Greco et al. used the OsteoSense™ 680EX dye to detect VSMC calcification in vitro [136]. They detected specific staining after 24 h of osteogenic stimulation [136]. The staining was also successfully applied in an ex vivo aortic ring calcification assay [136]. 

BoneTag probe is a tetracycline derivative coupled to IRDye 800CW [137]. The BoneTag probe has been used to detect the mineralization of osteoblast cultures in vitro with high signal intensities; however, further studies are needed to test whether it is useful in detecting VSMC calcification [137].

## 8. Using In Vitro Calcification Methods in Drug Discovery 

Based on our own and other research groups’ results, cellular models are very useful for determining whether a particular substance enhances or potentially inhibits calcification. For example, using HAoSMC and VIC calcification models, we found that Daprodustat, a prolyl hydroxylase inhibitor developed to treat anemia in CKD patients, accelerates Pi-induced calcification in both HAoSMCs and VICs [102,103]. Based on this result, we tested the effect of DPD in a mouse model of CKD. We found that DPD treatment corrected CKD-associated anemia; however, it accelerated arterial and valve calcification [102,103]. Another example is magnesium (Mg), which has been shown to inhibit BGP-induced VSMC calcification [52]. In agreement with the in vitro results, a study by Lindeers et al. found that dietary Mg supplementation attenuates abdominal aorta calcification in a partial nephrectomy rat model of CKD [138]. However, a recently reported clinical trial (MAGiCAL-CKD) concluded that Mg supplementation did not mitigate the progression of vascular calcification in CKD patients, despite a significant increase in plasma Mg levels [139]. 

## 9. Conclusions

Arterial calcification, a condition without preventive or treatment options, predicts future cardiac events and deaths. Therefore, further effort is needed to apprehend the mechanisms of vascular and aortic valve calcification. In vitro cell and ex vivo tissue culture models are the basic, necessary, and useful tools in calcification research. In many cases, the results obtained by using cell culture models are translatable to in vivo conditions. Besides basic science, these relatively cheap cell culture models can be used to screen compounds to identify novel calcification inhibitors and to investigate potential pro-calcifying side-effects of new drugs, especially if the target group is prone to calcification, such as patients with CKD or diabetes or elderly people.

## Figures and Tables

**Figure 1 biomedicines-12-02155-f001:**
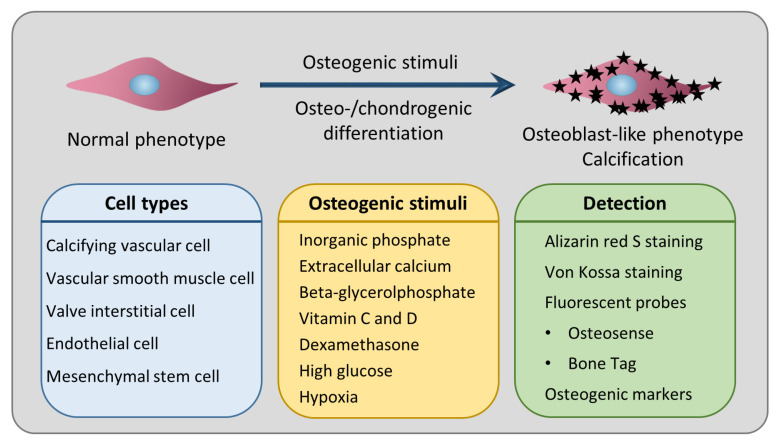
Graphical summary of the review.

## Data Availability

Not applicable.

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
