# Peer review of "In Vitro Models of Cardiovascular Calcification"

_biomedicines, 2024, doi:10.3390/biomedicines12092155_

Round 1

Reviewer 1 Report

Comments and Suggestions for Authors

Proposed review clearly describes models of cardiovascular calcification but some limitations are visible.

Vitamin C importance in cardiovascular calcification should be added together with Vitamine D to the Figure 1 and appriopriate places of the manuscript as in references [for example from papers like below:

Arakawa et al., J. Cardiovascular Pharmacology 2004, 42(6): 745-751

Aguirre and May,  Pharmacology & Therapeutics 2008, 119(1): 96-110].

Table 1 may be better constructed if HAoSMC, Bovine VSMC and VIC cell types will be joined in first column.

OPN abbreviation of osteopontin may be used in all manuscript.

Description of the endothelial cells (EC) and relevant references should be indicated in the Table 1.

ARS staining is extracted also by cetylpyridinium chloride (CPC).

Author Response

We thank the reviewer for the critical review and the helpful suggestions. We would like to take the opportunity to respond point by point to the reviewer’s specific comments.

Comment 1

Proposed review clearly describes models of cardiovascular calcification but some limitations are visible. Vitamin C importance in cardiovascular calcification should be added together with Vitamine D to the Figure 1 and appropriate places of the manuscript as in references [for example from papers like below:

Arakawa et al., J. Cardiovascular Pharmacology 2004, 42(6): 745-751

 Aguirre and May, Pharmacology & Therapeutics 2008, 119(1): 96-110].

Response

Thank you for the suggestion. We have completed the manuscript with vitamin C (section 6.4, page 8), and cited the relevant papers suggested by the reviewer.

Comment 2

Table 1 may be better constructed if HAoSMC, Bovine VSMC and VIC cell types will be joined in first column.

Response

Thank you for the comment. We reconstructed the table as suggested.

Comment 3

OPN abbreviation of osteopontin may be used in all manuscript.

Response

As suggested we use OPN to abbreviate osteopontin.  

Comment 4

Description of the endothelial cells (EC) and relevant references should be indicated in the Table 1.

Response

We included ECs in the revised table.

Comment 5

ARS staining is extracted also by cetylpyridinium chloride (CPC).

Response

Thank you for the comment. CPC is the other name of hexadecyl pyridinium chloride.

Reviewer 2 Report

Comments and Suggestions for Authors

The review discusses the current state of the problem of studying the factors and mechanisms of vascular calcification in experimental cell models. The authors analyzed a sufficient number of literature sources, including those from the last 5 years. The article is very interesting for scientists. The article requires correction.

My comments:

1. Almost the entire review is devoted to discussing models for studying calcification in vitro. Ex vivo models are described only in 2 subsections (lines 203-225). Therefore, the term ex vivo should be removed from the review title.

2. The text with the results of our own research (lines 344-348) from the Conclusion section should be removed, since it is superfluous in this section. This text should be slightly expanded and inserted into a separate subsection in the text of the review.

Author Response

Reviewer #2

We thank the reviewer for the critical review and the helpful suggestions. We are taking the opportunity to respond point by point to the reviewer’s comments.

Comment 1

Almost the entire review is devoted to discussing models for studying calcification in vitro. Ex vivo models are described only in 2 subsections (lines 203-225). Therefore, the term ex vivo should be removed from the review title.

Response

We changed the title as suggested.

Comment 2

The text with the results of our own research (lines 344-348) from the Conclusion section should be removed, since it is superfluous in this section. This text should be slightly expanded and inserted into a separate subsection in the text of the review.

Response

Thank you for the comment. The suggestion is taken; we wrote a new section: 8 Using the in vitro calcification methods in drug discovery. We moved our research from the conclusions to the new section and completed it with other relevant studies.

Reviewer 3 Report

Comments and Suggestions for Authors

Dear Editor,

I have read with extreme interest the paper by Dr Roth et al. entitled “In vitro and ex vivo models for cardiovascular calcification”.

For research and clinical purposes the argument is a hot topic in the field of translational cardiovascular medicine, mainly due to the clinical implications and possible drug treatments options. Even MDPI has very recently published a very excellent review on BIOLOGY (doi: 10.3390/biology13080565. Converging Mechanisms of Vascular and Cartilaginous Calcification).

Having this recently published review in mind, I have the following considerations:

1. Both the genes NOTCH1 and BMPR2 are proven to be associated to vessel calcification (mainly in valves) in different genetic entities. In the review by Dr Toth et al. they are not covered  at all and this -in my perspectie) is a vulnus; 2. General Arterial Calcification in Infancy (GACI) is a rare conditions in which a generalised vascular calcification is observed. The associted gene is ENNP1. Once agin thi is not mentioned.

In other words, I would include in the review, genetic conditions that are associated with the phenotype presented in this paper as model for ex vivo research also because some therapeutical approch is in fieri.

I am reminding thta Loss of Function (LoF) mutations in the PCSK9 gene led to the discovery of monoclonal antibodies for Familial Hypecholesterolemia (FH).

Author Response

Reviewer #3

We thank the reviewer for the critical review and the helpful suggestions. We are taking the opportunity to respond point by point to the reviewer’s comments.

Comment 1

  1. Both the genes NOTCH1 and BMPR2 are proven to be associated with vessel calcification (mainly in valves) in different genetic entities. In the review by Dr Toth et al. they are not covered at all and this -in my perspective) is a vulnus;

Response

Thank you for the comment. We completed the part 3. Osteo-/chondrogenic differentiation as the underlying cellular mechanism of calcification; and in the revised version we mentioned both BMP2 and NOTCH pathways.

Comment 2

General Arterial Calcification in Infancy (GACI) is a rare condition in which a generalized vascular calcification is observed. The associated gene is ENNP1. Once again this is not mentioned. In other words, I would include in the review, genetic conditions that are associated with the phenotype presented in this paper as a model for ex vivo research also because some therapeutical approaches is in fieri.

Response

Thank you for the comment. We have added a new paragraph to the introduction (paragraph 2). In this part, we described those rare genetic mutations that exhibit a cardiovascular calcification phenotype. We also completed section 5 Ex vivo models of cardiovascular calcification with some corresponding thoughts.

Round 2

Reviewer 3 Report

Comments and Suggestions for Authors

Dear Editor,

I have read the revised version of the manuscript that in the present form has addressed all of my points. The present paper is enormously improved from the original version.